# Nosocomial Vs. Community-Acquired Infective Endocarditis in Spain: Location, Trends, Clinical Presentation, Etiology, and Survival in the 21st Century

**DOI:** 10.3390/jcm8101755

**Published:** 2019-10-22

**Authors:** Christian Ortega-Loubon, María Fe Muñoz-Moreno, Irene Andrés García, Francisco Javier Álvarez, Esther Gómez-Sánchez, Juan Bustamante-Munguira, Mario Lorenzo-López, Álvaro Tamayo-Velasco, Pablo Jorge-Monjas, Salvador Resino, Eduardo Tamayo, María Heredia-Rodríguez

**Affiliations:** 1Department of Cardiac Surgery, Clinic University Hospital of Valladolid, Ramon y Cajal Ave. 3, 47003 Valladolid, Spain; 2BioCritic. Group for Biomedical Research in Critical Care Medicine, Ramon y Cajal Ave. 7, 47005 Valladolid, Spainmariolorenzo17@yahoo.es (M.L.-L.);; 3Unit of Research, Clinic University Hospital of Valladolid, Ramon y Cajal Ave. 3, 47003 Valladolid, Spain; 4Department of Preventive Medicine and Public Health, Clinic University Hospital of Valladolid, Ramon y Cajal Ave. 3, 47003 Valladolid, Spain; 5Department of Pharmacology and Therapeutics, Faculty of Medicine, University of Valladolid, Ramon y Cajal Ave. 7, 47005 Valladolid, Spain; 6Department of Anaesthesiology, Clinic University Hospital of Valladolid, Ramon y Cajal Ave. 3, 47003 Valladolid, Spain; 7Department of Surgery, Faculty of Medicine, University of Valladolid, Ramon y Cajal Ave 7, 47005 Valladolid, Spain; 8Department of Hematology, Clinic University Hospital of Valladolid, Ramon y Cajal Ave. 3, 47003 Valladolid, Spain; 9Unit of Infection and Immunity, National Center of Microbiology, Instituto de Salud Carlos III, Pozuelo Road 28, 28222 Majadahonda, Madrid, Spain

**Keywords:** infective endocarditis, nosocomial, community-acquired, mortality, location

## Abstract

Major changes have occurred in the epidemiology and etiology of infective endocarditis (IE). Nevertheless, the differences between nosocomial infective endocarditis (NIE) and community-acquired infective endocarditis (CIE) have not been addressed in a population-based study. We conducted a retrospective, nationwide, temporal trend study from 1997 to 2014 analyzing the epidemiology, clinical, geographical, meteorological characteristics of patients diagnosed with IE in Spain, to distinguish NIE from CIE. Among 25,952 patients with IE (62.2 ± 18·6 years; 65.9% men), 45.9% had NIE. The incidence of IE increased from 2.83 to 3.73 due to the NIE incidence increment with a decline in CIE. Patients with NIE were older (63.8 years vs. 60.8 years, *p* < 0·001), presented a higher Charlson index (1.22 vs. 1.03, *p* < 0.001), a greater history of implanted cardiac devices (8.7% vs. 4.6%, *p* < 0.001), and higher mortality (31.5% vs. 21.7%, *p* < 0.001). The most frequent microorganism for both NIE and CIE was *Staphylococcus* (*p* < 0.001), and the North reported a higher incidence (*p* < 0.001). Risk factors of mortality for NIE were age, Charlson index, hemodialysis, shock, heart failure, and stroke. Risk factors for CIE included female sex, renal disease, and cardiac-device carriers. The etiology of IE shifted from community origins to mostly nosocomial-associated infections. Higher morbidity, mortality, and poorer outcomes are associated with NIE.

## 1. Introduction

Infective endocarditis (IE) has a low incidence but is associated with a high degree of morbidity and mortality despite adequate antimicrobial management and cardiac surgery. The epidemiology of the disease has changed since William Osler’s 1885 study established a standard for clinical and pathophysiological correlation [1,2]. Today we have a better understanding of IE’s epidemiology due to changes in the prevalence of risk factors and better diagnostic tools and treatment [3,4]. 

There has been a reduction in rheumatic heart disease cases due to *Streptococcus* spp., principally in younger women, and an increase in the incidence of acute staphylococcal endocarditis in older men [2,5]. Several factors have profoundly impacted the clinical spectrum of IE: hemodialysis, diabetes mellitus, intravenous drug use, human immunodeficiency virus, increased survival of the population at risk for IE, increased incidence of degenerative heart disease, interventional procedures, and increased use of intracardiac devices [6]. The update of the IE guideline published in 2007 and 2008 limited the use of antibiotic prophylaxis, recommending the cessation of antibiotic use in moderate-risk patients [7,8]. Therefore, it is important to determine the updated guidelines’ direct consequences on the overall incidence of IE, which have not yet been described in Spain. 

Depending on the mode of acquisition, IE is divided into community-acquired infective endocarditis (CIE) and nosocomial infective endocarditis (NIE), and everything suggests that with the shift in risk factors, the incidence of NIE is increasing. However, studies analyzing the differences between CIE and NIE are infrequent and come from isolated, small-population series and may not reflect real changes because the large population studies advocate IE as a whole. Furthermore, most studies consisted of single-center reports which limit the scope and statistical power needed to draw strong conclusions. The lack of nationwide studies has limited our understanding of trends and differences in IE. Furthermore, little is known about the influence of geographical and meteorological conditions in the incidence of bloodstream infections [9]. Toyoda et al. analyzed the difference between both types of IE [10]; however, no location differences or mortality predisposing risk factors were addressed. In addition, two isolated regions in the United States were analyzed but did not reflect the nation as a whole, and the environments of those studies differ from the environmental conditions in Europe.

Our study aimed to describe the global trend of IE in Spain, compare the characteristics between NIE and CIE, and determine their mortality associated risk factors.

## 2. Materials and Methods

### 2.1. Study Design, Data Source, and Case Identification

A nationwide retrospective study was carried out, including all patients admitted with IE in all hospitals in Spain between 1 January 1997 and 31 December 2014.

The minimum basic data set (MBDS) of the National Hospital Data Surveillance System in Spain, provided by the Ministry of Health of Spain, is one of the databases of hospitalized patients and the most important source of morbidity data, containing approximately 92% of all acute care hospital information in Spain. It provides an encrypted patient identification number, sex, birth date, hospital admission and discharge dates, medical center, diagnosis, and procedure codes according to the International Classification of Diseases 9th Revision, Clinical Modification (ICD-9-CM), and outcomes after discharge. Confidentiality was adequately protected according to the Spanish Data protection law. The Spanish Ministry of Health confirmed that our study fulfilled all ethical considerations according to Spanish legislation. The data collected from the MBDS were encoded to avoid duplication and to dissociate any information that might reveal the identity of the patients.

Patients were included in the study if they had their first episode of IE identified by either a primary or secondary diagnostic *ICD-9-CM* code 421.0, 421.1, 421.9, 112.81, 115.04, 115.14, or 115.94. To identify a cohort of incident cases and prevent double-counting of patients, the index episode (date admission exclusion) for IE from 1997 to 2014 was selected. Hospital admissions without a unique identifier, as well as readmissions, were excluded (Figure 1).

### 2.2. Study Variables

Demographic data included age, sex, locations, seasons, Charlson index (used to determine overall systemic health [11]), comorbidities, predisposing factors, acquisition mode of IE, causative microorganism, type of admission, intervention, length of stay, overall and 90-day mortality. The ICD-9 CM codes are listed in Appendix A. 

### 2.3. Definitions

NIE was defined as IE associated with medical interventions performed in hospital within the 8-week period before the onset of the disease, and⁄or IE occurring >48 h after admission, while all other episodes were defined as CIE [12]. 

### 2.4. Statistical Analysis

The crude incidence of IE was calculated by dividing the number of patients with the first episodes of IE in each year by Spain census populations in the same year facilitated by the Statistics National Institute reported per 100,000 habitants. Multivariable Poisson regression analysis was performed to evaluate temporal trends in the incidence of IE adjusting for age and sex. Trends in the incidence were estimated by the annual percentage change (ACP) with a 95% confidence interval (CI).

For descriptive analysis, continuous variables were reported as means with standard deviations, and categorical variables as percentages per the total number of IE cases. For 90-day mortality, the study cohort from 1 January 1997 through 31 December 2014, was used. Survival curves were drawn using the Kaplan–Meier method and compared using the log-rank test. Variables were included in univariate Cox regression analysis to evaluate the trend in mortality during the study period and risk factors of mortality. Multivariable Cox regression was performed, adjusting for age, sex, baseline comorbidities (i.e., hypertension, diabetes, renal disease, coronary artery disease, peripheral vascular disease, chronic pulmonary obstructive disease, liver disease, history of malignancy, and history of congenital heart disease), disease type, and acquisition mode.

All tests were 2-tailed. Hazard ratio (HR) with 95% CI and *p*-values were reported. The level of significance was set at *p*  <  0.05. Data were analyzed using IBM SPSS Statistics for Windows version 24·0 software (IBM Corp, Armonk, NY). 

## 3. Results

### 3.1. Incidence

A total of 25,952 cases of IE were diagnosed between 1997 and 2014 (mean age, 62.2 ± 18.6 years; 65.9% men). Among them, 11,921 (45.9%) cases were diagnosed as NIE and 14,031 (54.1%) as CIE. The crude incidence increased from 2.83 to 3.73 cases per 100,000 habitants annually (Annual percentage change 2.25; 95% CI, 0.66% to 3.86%; unadjusted Poisson regression, *p* = 0.724). The country’s geography was divided according to climate into North, Center, and South regions. Even though more IE cases were reported primarily in southerly locations, its overall incidence rates were almost doubled in the North compared to other regions (*p* < 0.001; Figure 2; Appendix A). The incidence of IE stratified by age and sex is depicted in Figure 3A, which shows a higher male patient proportion throughout the entire studied period (*p* < 0.001). Incidence was significantly higher in elderly patients (Appendix A).

### 3.2. Population Description

The epidemiological and clinical patient characteristics are shown in Table 1. Patients with IE in the latter period of the study were older, more likely to have higher Charlson index, hypertension, heart failure, chronic obstructive pulmonary disease, both moderate and severe diabetes with chronic complications, cancer, and required more hemodialysis compared to patients from earlier periods (Table 1).

From 1997 through 2014, the proportion of patients with a history of valve surgery increased from 8.9% to 11.1%, and the proportion of patients with implanted pacemakers or defibrillators certainly doubled from 4.0% to 8.3%. As a result, the proportion of patients with cardiac device-related endocarditis significantly increased from 11.2% to 27.4%, whereas the percentage of IE in drug users decreased (Table 1).

### 3.3. Evolution and Tendencies of IE

Overall, IE increased between 1997 and 2014, especially among older adults (Figure 3B). While the proportion of CIE accounted for more than half of the cases at the beginning of the study period with 59.9% in 1997, it dropped moderately to 48.8% by 2014 (*p* < 0.001). The percentage of NIE increased significantly from 40.1% in 1997 to 52% in 2014 (*p* < 0.001; Figure 4).

The maximum age-adjusted incidence of IE (events per 100,000 habitants) was observed in those older than 80 years, being higher in men for all age groups (Figure 3B). The proportion of patients who underwent cardiac surgery during their admission considerably grew in the analyzed period, from 18.7% to 24.6% (*p* < 0.001).

Regarding microorganisms, the incidence of all pathogens consistently increased over the years. IE was most frequently caused by Gram-positive (2070, 8.0%) followed by the Gram-negative (1443, 5.6%), fungi (159, 0.6%), and anaerobes (21, 0.1%). The incidence of IE caused by Gram-positive microorganisms increased from 5.1% to 8.1% per 100,000 inhabitants during the study period (*p* < 0.001).

### 3.4. Comparison between NIE and CIE

The clinical characteristics of NIE compared with CIE are summarized in Table 2. Patients with NIE were older (63.8 vs. 60.8; *p* < 0.001), presented higher Charlson index (1.22 vs. 1.03; *p* < 0.001), had a higher incidence of history of implanted cardiac devices (8.7% vs. 4.6%; *p* < 0.001) or valve replacements (11.3% vs. 8.2%; *p* < 0.001), and higher 90-day mortality (31.5% vs. 21.7%; *p* < 0.001). 

### 3.5. Mortality and Survival

Both overall and 90-day mortality increased from 1997 through 2014 from 22.3% to 29.8% and from 26.2% to 28.4%, respectively (*p* < 0.001). By age group, the percentage of deaths from IE was 11.1% in patients younger than 19 years, which progressively increased from 11.5% in the 30 to 34-year age group to 21.1% in the 40 to 59-year age group subjects, peaking to 31.3% in those older than 80 years. In terms of gender, mortality was higher in women than in men throughout all periods (30.4% vs. 25.4%; *p* < 0.001; Table 3). 

Concerning survival rates, compared to male patients, survival appeared to be lower for female patients (Figure 5A). Similarly, elderly patients presented the lowest survival (Figure 5B). Figure 5C displays the survival of patients with IE stratified by pathogens. Mortality was higher for Gram-negative microorganisms (HR, 2.68; 95% CI, 2.49–2.88), and Gram-positive infections (HR, 2.08; 95% CI, 1.94–2.22). Mortality was higher in NIE patients, gradually increasing from 27.4% in 1997 to 35.5% in 2014 (*p* < 0.001), and was associated with significantly higher mortality and poorer outcomes than CIE (HR, 1.43; 95% CI, 1.36–1.50; *p* < 0.001; Figure 5D).

### 3.6. Risk Factors of Mortality for NIE and CIE

Univariate Cox regression analysis for mortality associated with NIE and CIE is shown in Table 4. 

Multivariable Cox regression analysis identified age (HR, 1.02; 95% CI, 1.01–1.02), Charlson index (HR, 1.09; 95% CI, 1.07–1.12), hemodialysis (HR, 1.20; 95% CI, 1.08–1.34), shock (HR, 3.19; 95% CI, 2.98–3.42), heart failure (HR, 1.32; 95% CI, 1.23–1.41), and stroke (HR, 1.76; 95% CI, 1.57–1.98) as independent risk factors of mortality for both NIE and CIE. Along with these findings, female sex (HR, 1.11; 95% CI, 1.03–1.20), renal disease (HR, 1.15; 95% CI, 1.01–1.33), and pacemaker or defibrillator carriers (HR, 1.18; 95% CI, 1.01–1.38) were also mortality-associated factors for CIE (Table 5). 

## 4. Discussion

We noted a constant increase in the incidence of IE in Spain, principally because of a dramatic rise in the proportion of NIE concurrent with a decline in the percentage of CIE. NIE showed higher morbidity than CIE, as elderly patients were the most affected group. The North reported the highest CIE incidence. Cox regression analysis revealed that age, Charlson index, hemodialysis, shock, heart failure, and stroke were independent risk factors of mortality for both NIE and CIE. In addition, the female sex, renal disease, and use of pacemakers or defibrillators were mortality-associated factors of CIE.

Numerous studies have confirmed a rising trend of IE rates over time [13,14]. Olmos et al. found a significant increase in the incidence of IE in Spain from 2003 to 2014. Similarly, Bor et al. reported the rising number of IE episodes is due to *Staphylococcus aureus* infections associated with cardiac devices and implants [15]. The growing incidence of NIE could be related to the increase in the prevalence of healthcare-associated interventional procedures in recent years (e.g., hemodialysis, catheterization, intravenous line placement, cardiac implantable electronic devices, prostheses). Furthermore, this probably explains why IE rose in those patients older than 80 years. Another factor explaining the consistently increasing pattern might be the changes in the prevention of IE. The American Heart Association in 2007 revised and changed the consensus for antibiotic prophylaxis, recommending the withdrawal of routine antibiotic prophylaxis for dental procedures in low-risk and moderate-risk patients and most other invasive procedures in all patients [8]. A similar trend occurred in the United Kingdom, where a substantial increase in IE cases was reported after the introduction of these guidelines in 2008 [16].

We identified 45.9% of all IE to be NIE, which is moderately higher than previous reports, accounting for 10% to 30% of all IE episodes. This finding might be related to the increment of invasive procedures, major incidence of degenerative valve disease in elderly patients, and longer intensive care unit and hospital stays. In our study, 24.5% of patients underwent cardiac surgery during admission, which is similar to what has been reported in population-based studies from Italy and the United States [17,18].

In terms of gender, a major predominance of men was observed in comparison with women throughout the study period. Likewise, Toyoda et al. found a similar trend in their investigation conducted in the United States in which patients with IE were more likely to be male throughout the study period [10]. In contrast with Toyoda et al., who described a decline in the cases of nosocomial endocarditis, our findings highlight the paramount importance of NIE, the incidence of which steeply rose despite all the coordinated efforts to reduce hospital-acquired infections through mandatory continuous reporting, regulated interventions, or evidence-based consensus recommendations [19]. 

Concerning predisposing factors, a higher Charlson index (reflecting more comorbidities such as diabetes mellitus, hemodialysis, coronary artery disease, peripheral artery disease, cancer, as well as major complications ranging from sepsis, shock, heart failure, and cerebrovascular disease) were present in patients with NIE. This rise in underlying diseases might be related to the aging of the population and explains why those adults with NIE, especially those older than age 80 years, depicted the lowest survival rates. 

Regarding the microbiological profile, Gram-positive pathogens are still the most frequent microorganisms associated with IE, as previously reported in other series [3,5,13,15,20,21,22]. These microorganisms often have a healthcare-associated origin, widely recognized as virulent and resistant organisms, and are related to high rates of complications and high mortality rates. Selton-Suty et al. demonstrated *Staphylococcus aureus* as the only factor associated with a higher risk of in-hospital mortality in healthcare-associated IE [23]. Compared with fungi, anaerobic, and streptococcal infection, mortality was higher for Gram-positive and Gram-negative pathogens, which differs from Toyoda et al.’s findings [10].

Spain’s climate varies across its geography, resulting in three broad regions: the North, where Atlantic weather characterized by a large proportion of humid days and the prevalence of precipitation; the Center, distinguished by a continental plateau with less precipitation; and the South, which is characterized by a Mediterranean climate with almost no precipitation, higher temperatures, and longer, sunny days. The higher IE incidence in the North region may be due to the higher level of humidity experienced there compared to the Center or South regions of Spain. These findings emphasized the importance of the site of acquisition. Indeed, to our knowledge, this is the first study to report a significant difference in IE rates in terms of location. This corresponds with Blanco et al.’s findings, who demonstrated that increasing humidity is associated with greater *Staphylococcus aureus* colonization [9]. Certainly, *Staphylococcus aureus* infections are more common in humid weather conditions [24]. Wang et al. reported a similar correlation between humidity and *Staphylococcus aureus* infections [6]. 

As far as mortality is concerned, risk factors for NIE mortality were increased age, higher Charlson index, and associated organ dysfunction, such as stroke, shock, heart failure, and renal replacement therapy. In addition to these risk factors, being female, having renal disease, and carrying a pacemaker or defibrillator were also independent risk factors for CIE mortality. These results are similar to previous studies, which also identified advanced age, female sex, *Staphylococcus aureus* infection, heart failure, septic shock, alterations in consciousness, delay of appropriate antibiotics, and persistent bacteremia as independent risk factors of in-hospital mortality [5,14,21,25].

Certainly, NIE and CIE should be considered separately because the outcomes are different [26]. NIE mortality remains considerably high [2,16]. Despite advances over the last century in diagnosis, medical and surgical treatment, mortality rates have not changed substantially in the previous 40 years [27]. The global and 90-day mortality in our study was 27.2% and 26.2%, respectively, which was slightly higher than that reported in the literature. The current in-hospital mortality rate for patients with IE is 15% to 20%, with one-year mortality approaching 40% [2]. Moreover, NIE carries a higher morbidity and mortality when compared to CIE (33.9% vs. 22.4%, *p* <·0.001) [12,28,29]. NIE is associated with poor outcomes, which aligns with previous studies [20,21,30]. Moreover, older adults yielded higher mortality rates. This correlates with Bikdeli et al., who demonstrated that patients older than 85 years had higher rates of hospitalization and mortality compared with those in other age groups [6]. Women depicted lower survival rates compared to men in our study, which aligns with the findings of Giannitsioti et al., who identified that women with *Staphylococcus* bacteremia experienced increased mortality than male patients [12]. Given the available medical and surgical treatment limitations, it is unlikely that any further improvement in the outcome of IE will occur if no further progress is made in its prevention.

### Study Limitations

Even though this study has a national scope with a widely used, well-characterized database, it has some limitations. This is a retrospective study, the accuracy of which relies on adequate hospital coding. Thus, it is prone to a possible underestimation of the real number of cases and misclassification. In addition, our database does not regard antibiotic management of IE. Despite these limitations, this large cohort reflects the changing trends in IE from a national perspective. 

## 5. Conclusions

The overall incidence of IE in Spain gradually increased during the study period, given the consistently increasing NIE incidence along with a decline in the incidence of CIE. The clinical profile of NIE was different from that of CIE in many respects, including the presence of frailer, older adults with major comorbidities subjected to more invasive procedures. NIE patients were associated with a higher mortality rate and poorer prognosis than CIE patients. The aging population, underlying diseases, and staphylococcal infections may explain the rise of NIE incidence and its mortality pattern.

## Figures and Tables

**Figure 1 jcm-08-01755-f001:**
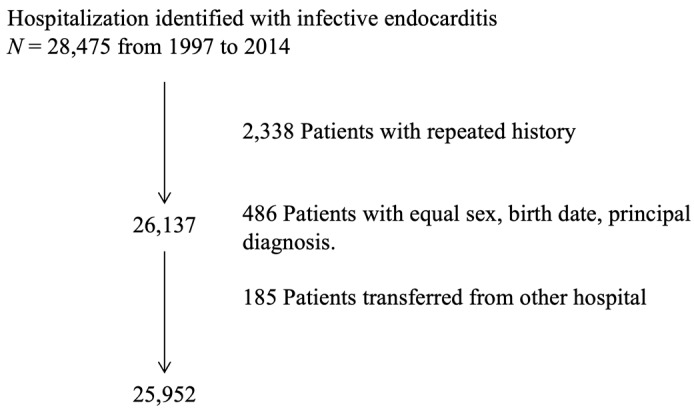
Flow chart of patient enrolment. Exclusion criteria.

**Figure 2 jcm-08-01755-f002:**
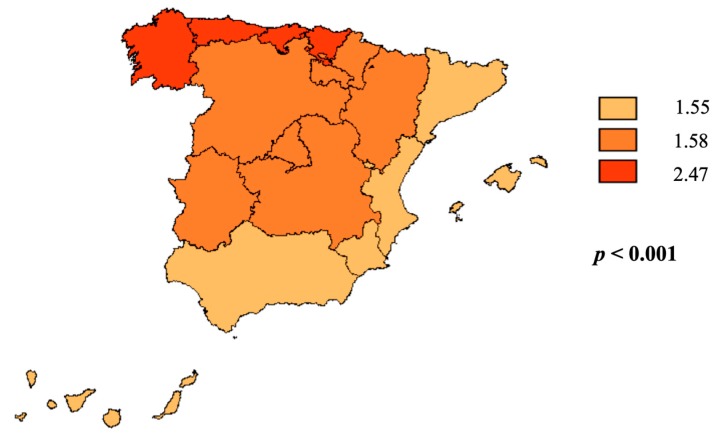
Incidence of infective endocarditis classified by location in Spain from 1997 to 2014. The units of measure for incidence are cases per 100,000 habitants. North: Atlantic weather characterized by a large proportion of humidity and precipitations. Center: Continental plateau with fewer precipitations. South: the Mediterranean with almost no precipitations, higher temperature, and longer sunny days.

**Figure 3 jcm-08-01755-f003:**
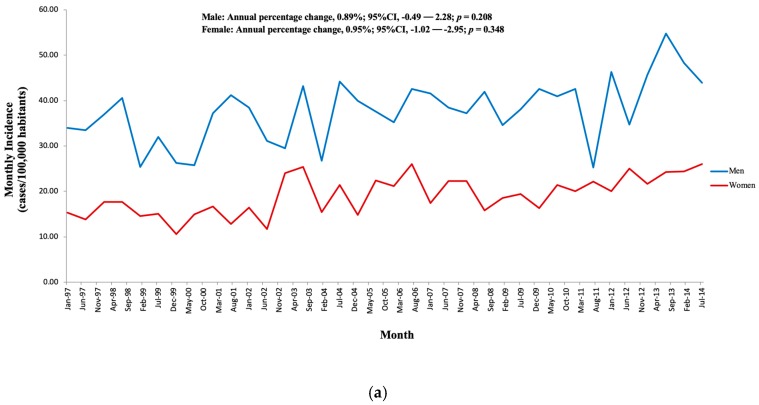
Incidence of infective endocarditis according to gender (**a**) and age groups (**b**) from 1997 to 2014 in Spain.

**Figure 4 jcm-08-01755-f004:**
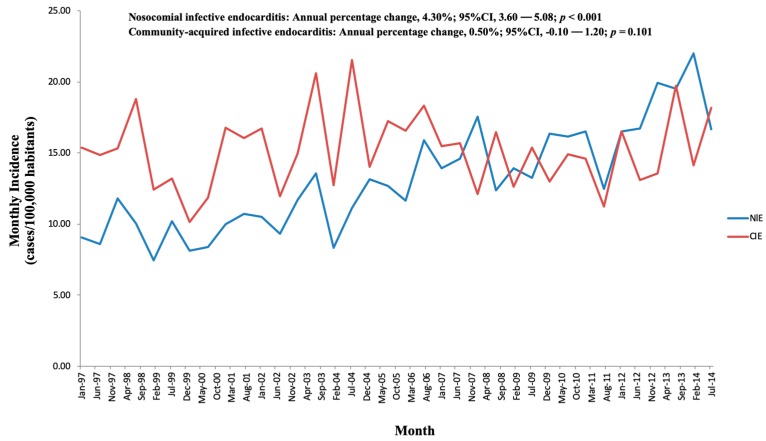
Incidence of infective endocarditis classified by nosocomial infective endocarditis and community-acquired infective endocarditis from 1997 to 2014 in Spain.

**Figure 5 jcm-08-01755-f005:**
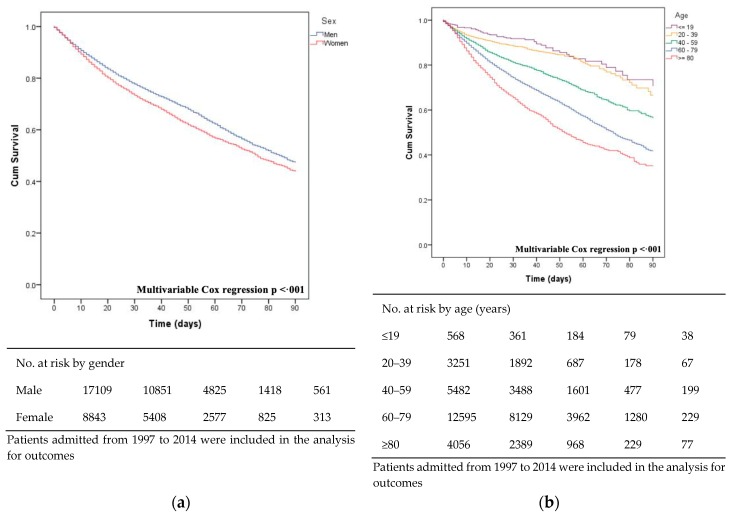
Survival of patients with infective endocarditis stratified by gender (**a**), age groups (**b**), causative microorganisms (**c**), and mode of acquisition (**d**) in Spain.

**Table 1 jcm-08-01755-t001:** Patient characteristics of infective endocarditis in Spain from 1997 to 2014. Overall and trends by year.

Variable		Global (*N* = 25,952)	1997–1999 (*n* = 3385)	2000–2004 (*n* = 6151)	2005–2009 (*n* = 7651)	2010–2014 (*n* = 8785)	*p*-Value
Incidence		3.28	2.83	2.94	3.36	3.73	0.724
Sex (% male)		17,109 (65.9)	2319 (68.5)	4102 (66.7)	4942 (64.8)	5746 (65.4)	0.002
Mean age in years (±SD)		62.2 (18.6)	53.6 (20.1)	59.8 (18.5)	62.8 (18)	66.6 (16.9)	<0.001
Age Group	≤19	568 (2.2)	90 (2.7)	127 (2.1)	170 (2.2)	181 (2.1)	<0.001
	20–39	3251 (12.5)	966 (28.5)	1000 (16.3)	799 (10.5)	486 (5.5)	
	40–59	5482 (21.1)	707 (20.9)	1372 (22.3)	1699 (22.3)	1704 (19.4)	
	60–79	12595 (48.5)	1376 (40.6)	3000 (48.8)	3790 (49.7)	4429 (50.4)	
	≥80	4056 (15.6)	246 (7.3)	652 (10.6)	1173 (15.4)	1985 (22.6)	
Seasons	Spring	6673 (25.7)	845 (25.0)	1626 (26.4)	1987 (26.0)	2215 (25.2)	0.244
	Summer	6539 (25.2)	907 (26.8)	1564 (25.4)	1889 (24.8)	2179 (24.8)	
	Autumn	6142 (23.7)	787 (23.2)	1438 (23.4)	1819 (23.8)	2098 (23.9)	
	Winter	6598 (25.4)	846 (25.0)	1523 (24.8)	1936 (25.4)	2293 (26.1)	
Locations	North	5489 (22.0)	646 (24.0)	1351 (22.5)	1613 (21.4)	1879 (21.6)	<0.001
	Center	7498 (30.1)	941 (35.0)	1692 (28.2)	2186 (29.0)	2679 (30.8)	
	South	11935 (47.9)	1104 (41.0)	2951 (49.2)	3737 (49.6)	4143 (47.6)	
Comorbidities							
Charlson Index Score (±SD)		1.1 (1.4)	0.7 (1.1)	1 (1.4)	1.2 (1.4)	1.3 (1.4)	<0.001
	0	10,825 (41.7)	2000 (59.1)	2823 (45.9)	3027 (39.7)	2975 (33.9)	<0.001
Charlson	1	7761 (29.9)	795 (23.5)	1711 (27.8)	2310 (30.3)	2945 (33.5)	<0.001
	2	4151 (16)	344 (10.2)	871 (14.2)	1247 (16.3)	1689 (19.2)	<0.001
	≥3	3215 (12.4)	246 (7.3)	746 (12.1)	1047 (13.7)	1176 (13.4)	<0.001
Hypertension		6145 (23.7)	346 (10.2)	1158 (18.8)	1974 (25.9)	2667 (30.4)	<0.001
Diabetes		4396 (16.9)	282 (8.3)	883 (14.4)	1390 (18.2)	1841 (21)	<0.001
Mild to Moderate Diabetes		3478 (13.4)	233 (6.9)	709 (11.5)	1107 (14.5)	1429 (16.3)	<0.001
Diabetes with Chronic Complications		940 (3.6)	49 (1.4)	177 (2.9)	292 (3.8)	422 (4.8)	<0.001
Coronary Artery Disease		1904 (7.3)	132 (3.9)	374 (6.1)	620 (8.1)	778 (8.9)	<0.001
Peripheral Vascular Disease		1307 (5)	75 (2.2)	280 (4.6)	406 (5.3)	546 (6.2)	<0.001
Rheumatic Disease		415 (1.6)	27 (0.8)	83 (1.3)	139 (1.8)	166 (1.9)	<0.001
COPD		3537 (13.6)	253 (7.5)	708 (11.5)	1101 (14.4)	1475 (16.8)	<0.001
Renal Disease		1446 (5.6)	230 (6.8)	492 (8)	536 (7)	188 (2.1)	<0.001
Hemodialysis		1808 (7.0)	82 (2.4)	244 (4.0)	533 (7.0)	949 (10.8)	<0.001
Liver Disease		1175 (4.5)	109 (3.2)	230 (3.7)	385 (5)	451 (5.1)	< 0.001
Cancer		475 (1.8)	32 (0.9)	88 (1.4)	158 (2.1)	197 (2.2)	<0.001
Human Immunodeficiency Virus		1361 (5.2)	459 (13.6)	428 (7.0)	313 (4.1)	161 (1.8)	<0.001
Receiving Intravenous Therapy		1152 (4.5)	79 (2.3)	204 (3.3)	394 (5.2)	485 (5.5)	<0.001
Sepsis		3458 (13.3)	274 (8.1)	714 (11.6)	1115 (14.6)	1355 (15.4)	<0.001
Shock		2583 (10.0)	208 (6.1)	498 (8.1)	807 (10.6)	1070 (12.2)	<0.001
AMI		1231 (4.7)	89 (2.6)	313 (5.1)	427 (5.6)	402 (4.6)	0.002
HF		6139 (23.7)	497 (14.7)	1263 (20.5)	1821 (23.9)	2558 (29.1)	<0.001
Stroke		1099 (4.2)	111 (3.3)	267 (4.3)	298 (3.9)	423 (4.8)	0.001
Hemiplegia		441 (1.7)	46 (1.4)	87 (1.4)	96 (1.3)	212 (2.4)	<0.001
Dementia		369 (1.4)	28 (0.8)	93 (1.5)	103 (1.3)	145 (1.7)	0.005
Predisposing factor							
History of Congenital Heart Disease		822 (3.2)	68 (2.2)	167 (2.7)	256 (3.4)	331 (3.8)	<0.001
History of Valve Surgery		2502 (9.6)	300 (8.9)	488 (7.9)	743 (9.7)	971 (11.1)	<0.001
History of Implanted Pacemaker or Defibrillator		1681 (6.5)	134 (4.0)	321 (5.2)	501 (6.6)	725 (8.3)	<0.001
Disease Type							
Cardiac device-related endocarditis		5506 (21.2)	378 (11.2)	1050 (17.1)	1670 (21.9)	2408 (27.4)	<0.001
Drug abuse-related endocarditis		1757 (6.8)	560 (16.5)	550 (8.9)	457 (6.0)	190 (2.2)	<0.001

Number of Organ Failure	0	19,453 (75)	2888 (85.3)	4926 (80.1)	5627 (73.7)	6012 (68.4)	<0.001
	1	5022 (19.4)	417 (12.3)	972 (15.8)	1540 (20.2)	2093 (23.8)	<0.001
	2	1171 (4.5)	66 (1.9)	206 (3.3)	377 (4.9)	522 (5.9)	<0.001
	≥ 3	306 (1.2)	14 (0.4)	47 (0.8)	87 (1.1)	158 (1.8)	< 0.001
Mode of Acquisition							
NIE		11,921 (45.9)	1357 (40.1)	2503 (40.7)	3490 (45.7)	4571 (52.0)	<0.001
CIE		14,031 (54.1)	2028 (59.9)	3648 (59.3)	4141 (54.3)	4214 (48.0)	<0.001
Causative organism							
Staphylococcus		8487 (32.7)	1098 (32.4)	1987 (32.3)	2510 (32.9)	2892 (32.9)	0.834
Staphylococcus aureus		5099 (19.6)	621 (18.3)	1293 (21.0)	1541 (20.2)	1644 (18.7)	<0.001
Methicillin-resistant		997 (3.8)	61 (1.8)	297 (4.8)	226 (3.0)	413 (4.7)	<0.001
Methicillin-sensitive		4102 (15.8)	560 (16.5)	996 (16.2)	1315 (17.2)	1231 (14.0)	<0.001
Streptococcus		641 (2.5)	113 (3.3)	123 (2.0)	165 (2.2)	240 (2.7)	<0.001
Gram-negative bacilli		4150 (16.0)	244 (7.2)	763 (12.4)	1275 (15.7)	1868 (21.3)	<0.001
Anaerobes		21 (0.1)	2 (0.1)	4 (0.1)	4 (0.1)	11 (0.1)	0.181
Fungi		234 (0.9)	27 (0.8)	55 (0.9)	64 (0.8)	88 (1.0)	0.629
Unspecified		12,419 (47.8)	1091 (56.2)	3219 (52.3)	3633 (48.3)	3686 (42.0)	< 0.001
Admission	Urgent	21227 (81.8)	2813 (83.1)	5029 (81.8)	6214 (81.4)	7171 (81.6)	0.116
	Elective	4671 (18)	547 (16.2)	1111 (18.1)	1412 (18.5)	1601 (18.2)	0.028
	Others-unknown	54 (0.2)	25 (0.7)	11 (0.2)	5 (0.1)	13 (0.1)	<0.001
Intervention		6364 (24.5)	634 (18.7)	1581 (25.7)	1986 (26)	2163 (24.6)	<0.001
Re-admission		4605 (17.7)	491 (14.5)	975 (15.9)	1348 (17.7)	1791 (20.4)	<0.001
LOS		30.5 (24.8)	28.8 (21.0)	30.2 (26.1)	31.8 (25.9)	30.2 (24.2)	0.014
Mortality							
Global Mortality		6952 (27.2)	741 (22.3)	1560 (25.6)	2095 (27.7)	2556 (29.8)	<0.001
<90 days		6801 (26.2)	732 (21.6)	1530 (24.9)	2043 (26.8)	2496 (28.4)	<0.001

Values are expressed as the absolute number (percentage) or mean (standard deviation). Statistical significance was defined as *p* < 0.05. Abbreviations: AMI, acute myocardial infarction; CIE, community-acquired infective endocarditis; COPD, chronic obstructive pulmonary disease; HF, heart failure; LOS, length of stay; NIE, nosocomial infective endocarditis.

**Table 2 jcm-08-01755-t002:** Comparison between nosocomial infective endocarditis and community-acquired infective endocarditis in Spain from 1997 through 2014.

Variable		Total (*N* = 25,952)	NIE (*n* = 11,921)	CIE (*n* = 14,031)	*p*-Value
Sex (% male)		17,109 (65.9)	7623 (64.0)	9486 (67.6)	<0.001
Mean age in years (±SD)		62.2 ± 18.6	63.8 ±18.2	60.8 ± 18.7	<0.001
Age Group	≤19	566 (2.2)	298 (2.5)	270 (1.9)	<0.001
	20–39	3251 (12.5)	1166 (9.8)	2085 (14.9)	
	40–59	5482 (21.2)	2298 (19.3)	3184 (22.7)	
	60–79	12595 (48.5)	6144 (51.5)	6451 (46.0)	
	≥80	4056 (15.6)	2015 (16.9)	2041 (14.6)	
Seasons	Spring	6673 (25.7)	3041 (25.5)	2632 (25.9)	0.014
	Summer	6539 (25.2)	2908 (24.4)	3631 (25.9)	
	Autumn	6142 (23.7)	2871 (24.1)	3271 (23.3)	
	Winter	6598 (25.4)	3101 (26.0)	3497 (24.9)	
Location	North	5489 (22.0)	2581 (22.4)	2908 (21.7)	<0.001
	Center	7498 (30.1)	3647 (31.7)	3851 (28.7)	
	South	11935 (47.9)	5279 (45.9)	6656 (49.6)	
Comorbidities					
Charlson Index Score (±SD)		1.1 ± 1.4	1.22 ± 1.5	1.03 ± 1.3	<0.001
	0	10825 (41.7)	4496 (37.7)	6329 (45.1)	<0.001
Charlson	1	7761 (29.9)	3725 (31.2)	4036 (28.8)	
	2	4151 (16)	2051 (17.2)	2100 (15.0)	
	≥3	3215 (12.4)	1649 (13.8)	1566 (11.2)	
Hypertension		6145 (23.7%)	2862 (24.0)	3283 (23.4)	0.249
Diabetes		4396 (16.9%)	2137 (17.9)	2259 (16.1)	<0.001
Mild to Moderate Diabetes		3478 (13.4%)	1660 (13.9)	1818 (13.0)	0.023
Diabetes with Chronic Complications		940 (3.6%)	488 (4.1)	452 (3.2)	<0.001
Coronary Artery Disease		1904 (7.3)	1039 (8.7)	865 (6.2)	<0.001
Peripheral Vascular Disease		1307 (5)	676 (5.7)	631 (4.5)	<0.001
Rheumatic Disease		415 (1.6)	211 (1.8)	204 (1.4)	0.043
COPD		3537 (13.6)	1660 (13.9)	1877 (13.4)	0.200
Renal Disease		1446 (5.6)	654 (5.5)	792 (5.6)	0.579
Hemodialysis		1808 (7.0)	990 (8.3)	818 (5.8)	<0.001
Liver Disease		1175 (4.5)	518 (4.4)	657 (4.7)	0.193
Cancer		475 (1.8)	244 (2.0)	231 (1.6)	0.016
Human Immunodeficiency Virus		1361 (5.2)	629 (5.3)	732 (5.2)	0.831
Receiving Intravenous Therapy		1162 (4.5)	819 (6.9)	343 (2.4)	<0.001
Sepsis		3458 (13.3)	2183 (18.3)	1275 (9.1)	<0.001
Shock		2583 (10.0)	1524 (12.8)	1059 (7.5)	<0.001
AMI		1231 (4.7)	753 (6.3)	478 (3.4)	<0.001
HF		6139 (23.7)	2960 (24.8)	3179 (22.7)	<0.001
Stroke		1099 (4.2)	598 (5.0)	501 (3.6)	<0.001
Hemiplegia		441 (1.7)	251 (2.1)	190 (1.4)	<0.001
Dementia		369 (1.4)	186 (1.6)	183 (1.3)	0.008
Predisposing Factor					
History of Congenital Heart Disease		822 (3.2)	298 (2.5)	524 (3.7)	<0.001
History of Valve Surgery		2502 (9.6)	1350 (11.3)	1152 (8.2)	<0.001
History of Implanted Pacemaker or Defibrillator		1681 (6.5)	1039 (8.7)	642 (4.6)	<0.001
Disease Type					
Cardiac device-related endocarditis		5506 (21.2)	4702 (39.4)	804 (5.7)	<0.001
Drug abuse-related endocarditis		1757 (6.8)	579 (4.9)	1178 (8.4)	<0.001
Number of Organ Failure	0	19453 (75)	8504 (71.3)	10949 (78.0)	<0.001
	1	5022 (19.4)	2555 (21.4)	2467 (17.6)	<0.001
	2	1171 (4.5)	678 (5.7)	493 (3.5)	<0.001
	≥3	306 (1.2)	184 (1.5)	122 (0.9)	<0.001
Causative Organism					
*Staphylococcus*		8487 (32.7)	4277 (35.9)	4210 (30.0)	<0.001
*Staphylococcus aureus*		5099 (19.6)	2446 (20.5)	2653 (18.9)	<0.001
Methicillin-resistant		997 (3.8)	598 (5.0)	399 (2.8)	<0.001
Methicillin-sensitive		4102 (15.8)	2446 (20.5)	2653 (18.9)	<0.001
*Streptococcus*		641 (2.5)	373 (3.1)	268 (1.9)	<0.001
Gram-negative bacilli		4150 (16.0)	1807 (15.2)	2343 (16.7)	<0.001
*Anaerobes*		21 (0.1)	12 (0.1)	9 (0.1)	0.303
*Fungi*		234 (0.9)	168 (1.4)	66 (0.5)	<0.001
Unspecified		12,419 (47.8)	5284 (44.3)	7135 (50.8)	<0.001
Admission	Urgent	21227 (81.8)	9607 (80.6)	11620 (82.8)	<0.001
	Elective	4671 (18)	2289 (19.2)	2382 (17.0)	
	Others-unknown	54 (0.2)	25 (0.02)	29 (0.02)	
Intervention		6364 (24.5)	3357 (28.2)	3007 (21.4)	<0.001
Re-admission		4605 (17.7)	2393 (20.1)	2212 (15.8)	<0.001
LOS		30.5 (24.8)	31.3 (29.0)	30.0 (20.6)	<0.001
Mortality					
Global		6952 (27.2)	3868 (32.9)	3084 (22.4)	<0.001
<90 d		6801 (26.2)	3761 (31.5)	3040 (21.7)	<0.001

Values are expressed as the absolute number (percentage) or mean (standard deviation). Statistical significance was defined as *p* < 0.05. Abbreviations: AMI, acute myocardial infarction; CIE, community-acquired infective endocarditis; COPD, chronic obstructive pulmonary disease; HF, heart failure; LOS, length of stay; NIE, nosocomial infective endocarditis.

**Table 3 jcm-08-01755-t003:** Overall and trends of mortality in Spain from 1997 to 2014 classified by gender, age group, and mode of acquisition of infective endocarditis.

Variable		Global (*N* = 6952)	1997–1999 (*n* = 741)	2000–2004 (*n* = 1560)	2005–2009 (*n* = 2095)	2010–2014 (*n* = 2556)	*p*-Value
Sex	Men (%)	4302 (25.6)	457 (20.1)	974 (24.0)	1285 (26.3)	1586 (28.3)	<0.001
	Women (%)	2650 (30.4)	284 (27.0)	586 (28.9)	810 (30.4)	970 (32.6)	
Age Group	≤19	63 (11.1)	12 (13.5)	16 (12.6)	14 (8.2)	21 (11.6)	<0.001
	20–39	365 (11.5)	102 (10.8)	117 (11.8)	94 (12.0)	52 (11.1)	
	40–59	1139 (21.1)	141 (20.3)	271 (20.0)	358 (21.3)	369 (22.2)	
	60–79	3886 (31.3)	393 (28.9)	920 (31.0)	1196 (31.7)	1377 (31.8)	
	≥80	1499 (31.3)	93 (38.1)	236 (36.4)	433 (37.3)	737 (38.1)	
Mode of Acquisition	NIE	3868 (32.9)	367 (27.4)	745 (30.1)	1162 (33.5)	1594 (35.5)	<0.001
	CIE	3084 (22.4)	374 (18.8)	815 (22.5)	933 (22.8)	962 (23.5)	

Values are expressed as the absolute number (percentage). Statistical significance was defined as *p* < 0.05. Abbreviations: CIE, community-acquired infective endocarditis; NIE, nosocomial infective endocarditis.

**Table 4 jcm-08-01755-t004:** Overall and trends of mortality in Spain from 1997 to 2014 classified by gender, age group, and mode of acquisition of infective endocarditis.

Variable	NIE		CIE	
	HR (95% CI)	*p*-Value	HR (95% CI)	*p*-Value
Female sex	1.08 (1.01–1.15)	0.022	1.32 (1.22–1.42)	<0.001
Age	1.01 (1.01–1.02)	<0.001	1.02 (1.02–1.03)	<0.001
Pacemaker or defibrillator placement	0.98 (0.87–1.09)	0.675	1.28 (1.09–1.49)	0.002
Diabetes	1.17 (1.08–1.27)	<0.001	1.28 (1.17–1.40)	<0.001
COPD	1.23 (1.23–1.34)	<0.001	1.25 (1.13–1.37)	<0.001
Coronary artery disease	1.00 (0.90–1.13)	0.893	1.25 (1.10–1.42)	0.001
Renal disease	1.17 (1.02–1.33)	<0.001	1.47 (1.29–1.66)	<0.001
Hemodialysis	1.13 (1.17–1.45)	<0.001	1.68 (1.48–1.90)	<0.001
Shock	3.17 (2.96–3.40)	<0.001	5.29 (4.88–5.74)	<0.001
Charlson	1.14 (1.12–1.16)	<0.001	1.20 (1.18–1.23)	<0.001
Heart Failure	1.65 (1.55–1.77)	<0.001	1.94 (1.80–2.09)	<0.001
Stroke	1.74 (1.55–1.95)	<0.001	2.27 (1.99–2.59)	<0.001
Type of admission	0.71 (0.64–0.77)	<0.001	0.139 (0.20–0.99)	<0.001
Gram-positive	1.83 (1.68–1.98)	<0.001	2.05 (1.83–2.30)	<0.001
Gram-negative	2.16 (1.98–2.34)	<0.001	3.10 (2.76–3.45)	<0.001

Abbreviations: CI, Confidence Interval; CIE, community-acquired infective endocarditis; COPD, chronic obstructive pulmonary disease; HR, hazard ratio; NIE, nosocomial infective endocarditis.

**Table 5 jcm-08-01755-t005:** Multivariable logistic regression analysis for mortality associated with nosocomial infective endocarditis and community-acquired infective endocarditis.

Variable	NIE		CIE	
	HR (95% CI)	*p*-Value	HR (95% CI)	*p*-Value
Age	1.02 (1.01–1.02)	<0.001	1.02 (1.01–1.02)	<0.001
Charlson Index	1.09 (1.07–1.12)	<0.001	1.10 (1.07–1.13)	<0.001
Hemodialysis	1.20 (1.08–1.34)	0.001	1.33 (1.17–1.51)	0.001
Shock	3.19 (2.98–3.42)	<0.001	5.14 (4.73–5.58)	<0.001
Heart Failure	1.32 (1.23–1.41)	<0.001	1.40 (1.29–1.52)	<0.001
Stroke	1.76 (1.57–1.98)	<0.001	1.94 (1.69–2.22)	<0.001
Female sex			1.11 (1.03–1.20)	0.005
Renal Disease			1.15 (1.01–1.33)	0.049
Pacemaker or defibrillator implanted			1.18 (1.01–1.38)	0.040

Abbreviations: CI, Confidence Interval; CIE, community-acquired infective endocarditis; COPD, chronic obstructive pulmonary disease; HR, hazard ratio; NIE, nosocomial infective endocarditis.

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
