# Peer review of "Nosocomial Vs. Community-Acquired Infective Endocarditis in Spain: Location, Trends, Clinical Presentation, Etiology, and Survival in the 21st Century"

_jcm, 2019, doi:10.3390/jcm8101755_

Round 1

Reviewer 1 Report

The manuscript “Nosocomial Vs Community-acquired Infective 2 Endocarditis in Spain…” by Ortega-Loubon et al. is well developed epidemiological study of infective endocarditis in Spain. In my opinion, the manuscript is valuable, however some comments need to be addressed.

Major comments:

Abstract: The description of the results is not fully clear. After the sentence: “Patients with NIE were older …” I expect you compare NIE with CIE. Next you write: “The most frequent microorganism was gram-positive (p<.001) …” – That would mean that CIE is caused mostly by Gram-negative bacteria, which is not true. Please improve this sentence to be clear without necessity the reading of whole text. Table 1: Causative microorganism of IE was presented only for 3693 from 25952 of total cases (about 14%). What about the rest of cases? Is the etiological factor unknown? Could you explain inverse relationship between the number of IE cases presented in Fig. 2 and in Table 2. The most incidence rates per 100 000 habitants in North Spain (Fig. 2) versus the most IE cases (both NIE and CIE) in South Spain (Table 2). Add short explanation to the discussion. Discussion, line 277-280: The authors' intention in this fragment is incomprehensible. Why did you use “while” to opposite the information: “…S. aureus as the only factor associated with a higher risk of in-hospital mortality in healthcare-associated IE.[23] While fungi, streptococcus, and Gram-negative microorganisms (eg, Escherichia coli and Pseudomonas) were significantly related to NIE …” when both data concerning hospital-associated IE? Improve this fragment.

Minor comments:

Abstract and whole text: Improve Gram-positive, Gram-negative bacteria (start with a capital letter). Abstract: Consider a replacement of “predictors” on e.g. risk factors Introduction, line 50-52: Make two separate sentence: “pathophysiological correlation[1,2]; today we have…” Introduction and whole text: Improve the format of microbial names as follow: Streptococcus (italics, the dot after spp), Staphylococcus aureus (italics), Escherichia coli, Pseudomonas (italics) etc. The legends to Fig. 2: infective endocarditis write with a lowercase letter The description for Table 1 and Table 2: Improve as follow: “Values are expressed as absolute number (percentage) or mean (standard deviation)” since the data are presented in one selected format.

Author Response

The manuscript “Nosocomial Vs Community-acquired Infective 2 Endocarditis in Spain…” by Ortega-Loubon et al. is well developed epidemiological study of infective endocarditis in Spain. In my opinion, the manuscript is valuable, however some comments need to be addressed.

Major comments:

1.Abstract: The description of the results is not fully clear. After the sentence: “Patients with NIE were older …” I expect you compare NIE with CIE. Next you write: “The most frequent microorganism was gram-positive (p<.001) …” – That would mean that CIE is caused mostly by Gram-negative bacteria, which is not true. Please improve this sentence to be clear without necessity the reading of whole text.

Response to reviewer

Thanks for your suggestion.  We have followed your suggestion and we have improved the text in the abstract.

The most frequent microorganism for both NIE and CIE was Staphylococcus (p < 0.001).

2.Table 1: Causative microorganism of IE was presented only for 3693 from 25952 of total cases (about 14%). What about the rest of cases? Is the etiological factor unknown?

Response to reviewer

Thank you for the question.  We have re-calculated the causative microoganisms using the CIE-9 codification, and we have added the unspecified etiology as well to table 1 and table 2.

Table 1

Global

N = 25952

1997–1999

(n=3385)

2000–2004

(n=6151)

2005–2009

(n=7651)

2010–2014

(n=8785)

p-value

Causative organism

Staphylococcus

8487 (32.7)

1098 (32.4)

1987 (32.3)

2510 (32.9)

2892 (32.9)

0.834

Staphylococcus aureus

5099 (19.6)

621 (18.3)

1293 (21.0)

1541 (20.2)

1644 (18.7)

<.001

Methicillin-resistant

997 (3.8)

61 (1.8)

297 (4.8)

226 (3.0)

413 (4.7)

<.001

Methicillin-sensitive

4102 (15.8)

560 (16.5)

996 (16.2)

1315 (17.2)

1231 (14.0)

<.001

Streptococcus

641 (2.5)

113 (3.3)

123 (2.0)

165 (2.2)

240 (2.7)

<.001

Gram-negative bacilli

4150 (16.0)

244 (7.2)

763 (12.4)

1275 (15.7)

1868 (21.3)

<.001

Anaerobes

21 (0.1)

2 (0.1)

4 (0.1)

4 (0.1)

11 (0.1)

0.181

Fungi

234 (0.9)

27 (0.8)

55 (0.9)

64 (0.8)

88 (1.0)

0.629

Unspecified

12419 (47.8)

1091 (56.2)

3219 (52.3)

3633 (48.3)

3686 (42.0)

<.001

3.Could you explain inverse relationship between the number of IE cases presented in Fig. 2 and in Table 2. The most incidence rates per 100 000 habitants in North Spain (Fig. 2) versus the most IE cases (both NIE and CIE) in South Spain (Table 2).

Response to reviewer

Thank you for your observation. Table 2 shows the frequency, and Figure 2 the incidence. Despite that the South of Spain presents a higher frequency of IE, its overall incidence is lower compared to the North when considering the population, because there are more people in the South than in the North of Spain.

Add short explanation to the discussion. Discussion, line 277-280: The authors' intention in this fragment is incomprehensible. Why did you use “while” to opposite the information: “…S. aureus as the only factor associated with a higher risk of in-hospital mortality in healthcare-associated IE.[23] While fungi, streptococcus, and Gram-negative microorganisms (eg, Escherichia coli and Pseudomonas) were significantly related to NIE …” when both data concerning hospital-associated IE? Improve this fragment.

Response to reviewer

Thank you for your suggestion.  We have improved the meaning of that fragment in line 281-283, and we have changed it as follows.

Compared with fungi, anaerobic, and streptococcal infection, mortality was higher for Gram-positive and Gram-negative pathogens, which differs from Toyoda et al’s findings.

Minor comments:

1.Abstract and whole text: Improve Gram-positive, Gram-negative bacteria (start with a capital letter).

Response to reviewer

Thank you for the observation.  We have capitalized those terms as suggested.

2.Abstract: Consider a replacement of “predictors” on e.g. risk factors Introduction, line 50-52: Make two separate sentence: “pathophysiological correlation[1,2]; today we have…”

Response to reviewer

Thank you for the observation. We have replaced “predictors” for “risk factors”, and have split that sentence as suggested.

3.Introduction and whole text: Improve the format of microbial names as follow: Streptococcus(italics, the dot after spp), Staphylococcus aureus (italics), Escherichia coliPseudomonas (italics) etc.

Response to reviewer

Thank you for the suggestion. We have followed your recommendation, and added italics and dots were needed.

4.The legends to Fig. 2: infective endocarditis write with a lowercase letter.

Response to reviewer

Thank you for the observation.  We have changed infective endocarditis to lower letter in legends of Fig. 2.

5.The description for Table 1 and Table 2: Improve as follow: “Values are expressed as absolute number (percentage) or mean (standard deviation)” since the data are presented in one selected format.

Response to reviewer

Thank you for the suggestion.  We have changed the description for Table 1 and 2 as recommended.

Reviewer 2 Report

1.- Degree of interest: HIGH

The manuscript is of interest because the authors DO a retrospective analysis of Endocarditis in Spain, including a large number of patients (25.952) and variables to establish differences between nosocomials and community acquired infective endocarditis. They study a subject with high morbi-mortality and sometimes difficult diagnosis

2.- Language

The presentation, language and figures, despite providing great information and study variables are adequate, clear and educational.

3.- Results

The results of the characteristics of patients with IE in Spain are well expressed in the text, tables and figures, both the general analysis of endocarditis and the differences between nosocomial and community acquired infective endocarditis.

4.-DISCUSSION

The authors make a detailed discussion where they show their results in relation to morbi-mortality, risk factors, etiology and demographic data of the population, which are similar to those published in the literature by other authors

5.- BIBLIOGRAPHY

It is adequate and up-to-date.

Author Response

Comments and Suggestions for Authors

Reviewer 2

1.- Degree of interest: HIGH

The manuscript is of interest because the authors DO a retrospective analysis of Endocarditis in Spain, including a large number of patients (25.952) and variables to establish differences between nosocomials and community acquired infective endocarditis. They study a subject with high morbi-mortality and sometimes difficult diagnosis

Response to Reviewer

Thank you for your comments.

2.- Language

The presentation, language and figures, despite providing great information and study variables are adequate, clear and educational.

Response to Reviewer

Thank you.  We greatly appreciate your observations.

3.- Results

The results of the characteristics of patients with IE in Spain are well expressed in the text, tables and figures, both the general analysis of endocarditis and the differences between nosocomial and community acquired infective endocarditis.

Response to Reviewer

Thanks for your observations.  

4.-DISCUSSION

The authors make a detailed discussion where they show their results in relation to morbi-mortality, risk factors, etiology and demographic data of the population, which are similar to those published in the literature by other authors.

Response to Reviewer

Thank you for taking your time to review the discussion.  We greatly appreciate it.

5.- BIBLIOGRAPHY

It is adequate and up-to-date.

Response to Reviewer.  

Thank you for your appreciation.  Indeed, the references are up-to-date.